# Knowledge, Perception, and Acceptance of HPV Vaccination and Screening for Cervical Cancer among Saudi Females: A Cross-Sectional Study

**DOI:** 10.3390/vaccines11071188

**Published:** 2023-07-01

**Authors:** Khulud Ahmad Rezq, Maadiah Algamdi, Raghad Alanazi, Sarah Alanazi, Fatmah Alhujairy, Radwa Albalawi, Wafa Al-Zamaa

**Affiliations:** Nursing Department, Faculty of Applied Medical Sciences, University of Tabuk, Tabuk 47512, Saudi Arabia; ialghamdi@ut.edu.sa (M.A.); wafaalzama@gmail.com (W.A.-Z.)

**Keywords:** HPV vaccination, screening, knowledge, perception, Saudi Arabia

## Abstract

Cervical cancer (CC) ranks as the eighth most prevalent malignancy in Saudi Arabian women of all ages. This cross-sectional study intends to assess women’s knowledge, perceptions, and acceptance concerning HPV vaccination and CC screening in Saudi Arabia as well as the contributing variables to women’s screening and vaccine acceptability. Data were collected between 1 April 2023 and 30 May 2023 through online questionnaires. Out of 421 responses, 70% of the studied sample had fair knowledge, and 30% had good knowledge related to cervical cancer screening and vaccine. Only 1.9% of the participants had a positive perception, while 41% of the participants had a negative perception toward cervical cancer screening and vaccine. A total of 38% of the participants were fearful of the side effects of the vaccine, while 22.2% doubted the effectiveness of the vaccine. Acceptance was much better correlated with perception (Rho = 0.47) than with knowledge (Rho = 0.177). However, this correlation remains weak. We conclude that Saudi women still have a poor understanding or impression of cervical cancer and prevention methods. Fear of the vaccination’s negative effects and skepticism about the effectiveness of the vaccine/screening have recently been the most often-mentioned hurdles to vaccine uptake.

## 1. Introduction

The third most frequent type of cancer globally by incidence is cervical cancer. Cervical cancer is thought to be the greatest cause of mortality in developing countries where it affects 80% of cases [1]. Cervical cancer is the eighth most prevalent malignancy in Saudi Arabian women of all ages [2,3]. Furthermore, the HPV information center in Saudi Arabia reports that there are 358 new cases of cancer every year with a crude incidence rate of 2.81. Additionally, according to the organization, 179 people die each year from cervical cancer with a crude fatality rate of 1.22 [3,4]. The HPV vaccination program was initiated in 2008 in Saudi Arabia targeting girls aged 9 to 26 years and males between the ages of 9 and 21 years. This vaccine provides protection against HPV infections, a significant risk factor for cervical cancer. Furthermore, HPV is strongly associated with other malignancies, such as vaginal, vulvar, anal, and oropharyngeal cancers, highlighting the importance of the vaccine as a preventive measure [5]. Saudi Arabia offers several methods for cervical cancer screening. The Pap smear test, which involves collecting cervical cells and examining them under a microscope for any precancerous or cancerous changes, is the most commonly used screening test. Additionally, high-risk HPV testing can be used in conjunction with the Pap smear or as a primary screening tool. Typically, these screening methods are recommended for women aged 21 to 65 years. For women between the ages of 21 and 29 years, a cervical test and pap smear are recommended every three years, while women between the ages of 30 and 65 years should undergo a pap smear and HPV test every five years. The HPV vaccination schedule in Saudi Arabia is as follows: One- or two-dose schedule for the primary target of girls aged 9–14 years, one- or two-dose schedule for young women aged 15–20 years, and two doses with a 6-month interval for women older than 21 years [4].

Globally, cancer-related deaths and disabilities had an $895 billion economic effect in 2008 [6,7]. Early cancer identification can lessen the financial burden [8]. More readily available than only treating cancer are cancer screenings and immunizations [9]. For instance, the Pap smear test in Saudi Arabia is free in public hospitals and costs $50 or more in private hospitals. Contrarily, the price of treating cervical cancer can range from $14,001 to $30,000 during the first year of therapy [8].

In contrast to the global view, the incidence of cervical cancer is very low in Saudi Arabia, ranking number 12 among all cancers in females, and accounts for only 2.4% of all new cases [4] despite the lack of national screening programs. The incidence rate is extracted from the Saudi Cancer Registry (SCR), which is a population-based registry developed in 1992. It was established under the jurisdiction of the Ministry of Health and commenced reporting cancer cases on 1 January 1994. Although it relies on the collaboration of 500 governmental and private hospitals, physician’s offices, cancer treatment centers, and pathology laboratories located throughout the country, full coverage of all cancer cases cannot be ascertained. Nevertheless, in view of the lack of national screening programs, the actual reason for this low incidence is unknown. The closed society and standards of mores could reduce women’s exposure to HPV infection [10,11,12,13]. Although cervical cancer is both preventable and curable, due to the lack of accessible screening in Saudi Arabia, most cases present at advanced stages [14,15] that require extensive chemo-radiation therapy. This is due to the lack of a proper screening program [4].

There are three basic methods for preventing and controlling cervical cancer. The primary preventive measures put the prevention of HPV infection at the forefront [16,17]. The detection of cervical pre-cancer in women who are at risk is the main goal of the secondary preventive methods [18]. Cervical cancer treatment, which entails surgery, radiation, chemotherapy, and palliative care, is the focus of the tertiary preventive methods [10,19]. To obtain a high aim from the program, however, cervical cancer prevention necessitates health education, counseling, and control [19]. Acceptance of HPV vaccination and screening is linked to understanding and perception in the case of cervical cancer [11,12,13]. Therefore, it is crucial to understand the knowledge, perception, and acceptability among women who are the program’s target audience if a long-lasting program, including HPV vaccination and screening, is to be developed [14]. Due to the fact that it is a clear measure of well-being and preventative practices, it complements the significance of research in women’s understanding of cervical cancer. The goal of this study is to assess women’s knowledge, perceptions, and acceptance concerning HPV vaccination and cervical cancer screening in Saudi Arabia as well as the contributing variables to women’s screening and vaccine acceptability.

## 2. Materials and Methods

### 2.1. Research Design

A cross-sectional design was used to assess knowledge, perception, and acceptance regarding HPV vaccination and screening of cervical cancer among women in Saudi Arabia.

### 2.2. Respondents and Sampling

For this study, a convenience sampling approach was employed to select the participants. The target population consisted of women between the ages of 18 and 50 years living in Saudi Arabia. Women who met the inclusion criteria of being within the specified age range and living in Saudi Arabia were considered eligible for participation. Those who lived outside Saudi Arabia and illiterate women were excluded from the study. The sample size was determined by using a single population percentage formula. The following assumptions were made: a 95% confidence level, a 5% margin of error, and a percentage of women between the ages of 18 and 50 years. As a result, 385 women were chosen as the sample size. Online surveys were distributed to the respondents using email and social media platforms, including Twitter, WhatsApp, and Telegram.

### 2.3. Instrument Development and Measures

The online questionnaire used in this study was adapted from previous literature [15]. Experts in the nursing field examined the questionnaire for content validity. To improve the language and expressivity of the survey items, a pilot sample (*n* = 20) was used. The pilot sample was excluded from the studied sample. The final questionnaire’s completion time was predicted to be between 5 and 10 min. The questionnaire was initially written in English before being translated into Arabic. The survey was circulated and piloted in Arabic. The online survey contained five parts: Part 1: Socio-Demographic of Respondents, which includes age, marital status, education, social status, and region; Part 2: Cancer-Related Characteristics of Respondents, which includes eight questions with yes or no responses; Part 3: Knowledge Regarding Cervical Cancer, HPV Vaccination, and Cervical Cancer, which consists of 12 questions with 4-point Likert scale responses (1: strongly disagree; 2: disagree; 3: agree; 4: strongly agree); Part 4: Perception Regarding Cervical Cancer, HPV Vaccination, and Cervical Cancer, which consists of 11 questions with 4-point Likert scale responses (1: strongly disagree; 2: disagree; 3: agree; 4: strongly agree); and Part 5: Accepts to Take HPV Vaccine/Screening, which involves two questions (factors enhancing uptake of HPV vaccine/screening and barriers to uptake of HPV vaccine/screening). In this study, the instruments’ Cronbach’s alpha was 0.788, showing acceptable evidence of reliability.

### 2.4. Scoring System

In general, the participants’ knowledge was assessed with 12 items, perceptions were assessed using 11 items, and acceptance was assessed using two items. The items allocated to the knowledge and perceptions domains were collected on a four-point Likert scale ranging between 1 = Strongly disagree and 4 = Strongly agree. Before the score calculation, three items in the knowledge domain were reverse scored in order to indicate better knowledge levels with higher scores (items #4, #5, and #9). The knowledge score was then computed by summing up the relevant items, and the knowledge was categorized into “Poor knowledge”, “Fair knowledge”, and “Good knowledge” for scores of 13–23, 24–36, and >37, respectively. In addition, the perceptions of the participants were totaled to represent the overall score. Participants with positive perceptions scored 11–21, those with neutral perceptions scored 22–33, and those with negative perceptions scored 34–44.

### 2.5. Statistical Analysis

The statistical analysis was performed with SPSS Software (v.27.0.0) and R version 4.3.1. Categorical data were presented as frequency and percentage, and numerical data were expressed as the mean and standard deviation (SD) or as the median and inter-quartile range (IQR). The factors associated with knowledge and perception regarding cervical cancer were assessed using the Kendall–Theil Sen Siegel nonparametric linear regression, where the scores of knowledge or perceptions were used as dependent variables, and the demographic and cancer-related characteristics were used as independent variables. Statistical significance was considered at *p* < 0.05. Using Bonferroni correction, the adjusted alpha point should be 0.01 instead of 0.05. We carried out a correlation analysis to evaluate the presence of correlation between acceptance and perception or knowledge.

### 2.6. Ethical Consideration

The University of Tabuk’s ethics committee granted clearance for this study with the approval number (UT-200-61-2022). A consent form was then collected from the respondents who were informed that their identities would be kept entirely anonymous and that the researchers would uphold the participants’ privacy. Additionally, the questionnaire’s proprietors gave their permission for its usage in this study.

## 3. Results

### 3.1. Demographic and Cancer-Related Characteristics

Initially, we received 432 responses. However, we analyzed 421 records since 11 participants declined to participate. More than half of the respondents aged 18 to <30 years (51.8%) were married (54.9%) and had a bachelor’s degree (64.8%). Additionally, 38.7% of respondents were residing in the Northern region, and 32.1% of respondents were residing in the Southern region. More than one-third of the participants were housewives (39.0%) or students (37.5%). The majority of the participants had never heard about cervical cancer (78.9%) and had never heard about screening (65.6%). However, only 31.1% of the participants had ever heard about the HPV vaccine. Additionally, 8.3% of the sample under study were screened against CC, and 20.2% had a family member who had been screened. Furthermore, 10.9% of the participants had received the vaccine, and 12.6% had a family member who had been vaccinated (Table 1).

### 3.2. Participants’ Knowledge and the Associated Factors

The overall knowledge scores of the participants ranged between 20.0 and 46.0 with a median (IQR) value of 35.0 (33.0, 37.0, Table 2). Poor knowledge was reported among 1.2%, fair knowledge among 69.4%, and good knowledge among 29.5% of the participants (Figure 1). Based on the univariate analysis (Table 3), the knowledge scores were significantly higher among divorced women and widows as well as those who live in the Western region and who had heard about HPV vaccination and screening (*p* < 0.001). In addition, those who were vaccinated or had a family member screened against CC had a significantly higher knowledge (*p* < 0.001). However, women residing in the Central and Eastern regions as well as post-graduates and those who had a family history of cancer had significantly lower knowledge scores (*p* < 0.001) (Table 3).

### 3.3. Participants Perceptions and the Associated Factors

The range of the perception scores was 12.0 to 4.0, and the median (IQR) value was 33.0 (30.0, 36.0, Table 2). Only 1.9% of the respondents had a positive perception, 57.0% had a neutral perception, and 41.1% had a negative perception (Figure 1). The analysis of the associated factors revealed that higher perception scores were apparent among women who were aged between 40 and 50 years or who had heard about cervical cancer, HPV vaccination, and screening as well as those who were married (*p* < 0.001). Nevertheless, divorced women, widows, and the unemployed as well as those who were aged between 30 and 40 years, carried bachelor or postgraduate degrees, lived in the Central region, and had a family history of cancer had a lower perception (*p* < 0.001, Table 4).

### 3.4. Facilitators and Barriers to HPV Vaccine Uptake

The most common facilitators to HPV vaccine uptake included the high efficacy of the vaccine and screening (36.6%) followed by the safety of the vaccine and screening procedures (31.0%). In contrast, the most frequently reported barriers to vaccine uptake included the fear of the side effects of the vaccine (39.1%) and the doubts regarding the effectiveness of the vaccine/screening (22.2%, Figure 2).

### 3.5. Correlation between Knowledge, Perception, and Acceptance

Knowledge was significantly correlated to both perception (*p*-value < 0.001) and acceptance (*p*-value < 0.001). The association between knowledge and perception was weak (Rho = 0.318, Figure 3). There was almost no correlation between knowledge and acceptance (Rho = 0.177, Figure 4).

Perception was significantly correlated to both knowledge (*p*-value < 0.001) and acceptance (*p*-value < 0.001). The association between knowledge and perception was weak (Rho = 0.318). There was a better correlation between perception and acceptance (Rho = 0.476, Figure 5).

Overall, acceptance was much better correlated with perception (Rho = 0.476) than with knowledge (Rho = 0.177). However, this correlation remains weak (Table 5).

## 4. Discussion

The purpose of this study is to assess women’s knowledge, perceptions, and acceptance concerning HPV vaccination and cervical cancer screening as well as the contributing variables to women’s screening and vaccine acceptability. The current study found that, among Saudi Arabian women, 1.2% had poor knowledge about cervical cancer screening and HPV vaccination, 69.4% had fair knowledge, and 29.5% had good knowledge (Figure 1). Despite the fact that 78.9% of the participants had heard about cervical cancer, and 65.6% had heard of screening, just 31.1% of the participants had heard of the HPV vaccine, only 10.9% had received it, and 12.6% knew a family member who had received the vaccine (Table 1). Furthermore, higher knowledge scores were reported among divorced women and those who heard about the HPV vaccine and screening, while lower knowledge scores were reported among women who lived in the Central region. This result is congruent with research conducted in Riyadh, Saudi Arabia by Jradi and Bawazir [20] who reported that Saudi women had poor knowledge of cervical cancer screening because most women believed they were not at risk for developing cervical cancer and that there was no need for a screening test if no signs or symptoms were present; the results point to a lack of understanding of cervical cancer and the need for screening. Moreover, our results align with a study conducted in Saudi Arabia, which also highlighted a low level of awareness among women regarding the significance of Pap smear tests and that HPV is the main cause of cervical cancer and sexually transmitted diseases [21]. Although cervical cancer burden and mortality have been shown to be reduced with HPV vaccination alone or in combination with screening in a variety of settings, basic knowledge about the cause of cervical cancer is still lacking in some societies, such as Saudi Arabian women, where myths about cervical cancer and HPV are widespread. Most likely, cultural or educational differences are to blame for these beliefs [20].

This study found that the highest reported factors facilitating the uptake of the HPV vaccine were its high efficacy and screening with a percentage of 36.6% followed by safety considerations regarding the vaccine and screening procedures at 31.0%. On the other hand, the most commonly reported obstacles to vaccine uptake were the fear of vaccine side effects at 39.1% and doubts about the effectiveness of the vaccine or screening at 22.2% (Figure 2). These findings underscore the need for increased educational efforts aimed at addressing these concerns and encouraging vaccine uptake. Additionally, this study found that only 1.9% of the participants had a positive perception of cervical cancer and HPV vaccination or screening, while 57% had a neutral perception, and 41.1% had a negative one (Figure 1). Women who were aware of cervical cancer and had received HPV vaccination had higher perception scores (Table 3). These results suggest that improving Saudi women’s understanding of cervical cancer prevention and educating them about HPV infection’s significance and susceptibility are crucial. Our findings indicate that the more educational activities that are provided, the more positive the perception of HPV vaccination and cervical cancer screening becomes. Our findings were consistent with the findings of Alnafisah et al. [22] who reported negative attitudes towards screening but only modest awareness of cervical cancer among Saudi women in the Qassim region. These attitudes reflect local customs, feelings, and culture. Additionally, a study conducted in Riyadh, Saudi Arabia among 326 women further supports our findings. The results indicated that a limited proportion of women (32.8%) had awareness about the sexual transmission of HPV, while only 21% were knowledgeable about its connection to cervical cancer [23]. Therefore, the efforts to improve cervical cancer prevention and vaccination uptake in Saudi Arabia must consider cultural and societal factors in addition to providing education and awareness-raising initiatives.

Our study’s findings are consistent with other research indicating that Saudi women often have inadequate knowledge levels regarding cervical cancer and HPV vaccination. For example, Ali et al. [24] found that female students at Jouf University had a limited understanding of HPV vaccination and cervical cancer with embarrassment cited as the most common reason for screening rejection; in addition, HPV vaccination was viewed as a pointless procedure. Similarly, female college students in Hail, Saudi Arabia performed poorly on knowledge tests related to cervical cancer risk factors, symptoms, and preventative strategies, as reported by Altamimi [25]. These findings highlight the need for a greater focus on cervical cancer as a significant health issue affecting Saudi women. Cervical cancer is linked to various factors, including limited access to care, HPV incidence, a lack of structured screening programs, and social and cultural norms that promote specific attitudes and beliefs towards cervical cancer. Healthcare providers should prioritize health education efforts aimed at improving women’s attitudes and behaviors towards cervical cancer screening by assessing their perceptions and knowledge levels, as recommended by previous research [26]. HPV could potentially be associated with approximately 3% of all cancers [18], including vaginal, endometrial, and other cancers. According to the current study results, knowledge was found to be significantly correlated with both perception and acceptance. However, the correlation between knowledge and perception was weak, as demonstrated in Figure 3. Additionally, there was almost no correlation between knowledge and acceptance, as shown in Figure 4. On the other hand, perception was significantly correlated with both knowledge and acceptance with a slightly stronger correlation observed between perception and acceptance compared to perception and knowledge, as illustrated in Figure 5. Overall, the findings suggest that acceptance of the HPV vaccine is more strongly associated with perception than knowledge. However, it is important to note that even the correlation between acceptance and perception remains relatively weak, as indicated by Figure 5. These results highlight the need for interventions aimed at improving both knowledge and perception to enhance the acceptance of the HPV vaccine among Saudi women.

The findings of this study align with previous research studies that showed a positive association between knowledge of the HPV vaccine and vaccine acceptance. Gerend and Shepherd [27] found that college-aged women who had knowledge about HPV and the vaccine were more likely to accept the vaccine. Similarly, Wong et al. [28] reported that knowledge and attitude toward the HPV vaccine were positively associated with vaccine uptake among Chinese women. Reiter et al. [29] investigated the relationship between knowledge, perception, and acceptance of the HPV vaccine among parents of adolescent girls and found a positive association between knowledge and vaccine acceptance. Although there are existing studies that examined the correlation between knowledge and vaccine acceptance, the current study highlights the need for more research to understand the relationship between knowledge, perception, and acceptance of the HPV vaccine. Recent research conducted in Saudi Arabia on women’s understanding of cervical cancer consistently highlights the need for educational interventions to improve knowledge. One such intervention, described in El-Sayed et al.’s [30] study, utilized tele-nursing based on the Health Belief Model to significantly improve women’s beliefs and knowledge about preventing cervical cancer. Tele-nursing utilizes information and telecommunications technology to deliver, manage, and coordinate care and services, providing quick access to specialist knowledge and improving patient care. Our study supports the hypothesis that there is a strong correlation between knowledge, perception, and acceptance of cervical cancer, the HPV vaccine, and screening given the high levels of knowledge and acceptance among the participants who had heard about these topics. Further research is needed to explore this correlation in more depth.

### Strengths and Limitations

The included sample size in this study is an important point of strength and is consistent with the number of participants in similar studies. We managed to overcome the cultural obstacles preventing Saudi women from answering such a questionnaire by every possible means. We also sought every possible risk factor and association of acceptability with perception and knowledge. We attempted to add on previous studies by conducting a correlation analysis, which revealed that acceptance was much better correlated with perception than with knowledge. Although the sample was collected from multiple regions in Saudi Arabia, the use of a cross-sectional study design and convenience sampling may limit the generalizability of the findings to the broader population of women in Saudi Arabia. The sample obtained through convenience sampling may not be fully representative of the entire target population. Moreover, relying on self-administered online surveys may limit the researchers’ capacity to explore the participants’ responses in-depth or resolve any potential ambiguities that may arise.

## 5. Conclusions

Women in Saudi Arabia still have a negative perception about cervical cancer and its treatment options. Only 10.9% of the participants in the current study had ever received the HPV vaccine, and 12.6% had a family member who had received the vaccine. Furthermore, only 29.5% of the participants were well informed about cervical cancer, compared to 69.4% who were adequately informed. It was shown that there was a statistically significant correlation between women’s knowledge levels and marital status with knowledge scores being noticeably higher among divorced women. On the knowledge tests, however, women in the Central region performed noticeably worse. The most often cited barriers to vaccination uptake lately are worries about the adverse effects of the immunization and doubt about the effectiveness of the vaccine/screening.

### Recommendation and Implications

The current study’s findings highlight the need for a culturally appropriate public education program and awareness campaign on cervical cancer and its prevention in Saudi Arabia. It is crucial to provide all Saudi females with a comprehensive educational program to enhance their knowledge and health beliefs and encourage cervical cancer screening. The inclusion of the topic of cervical cancer in all-female educational settings is necessary. To fill in the existing knowledge gaps, media campaigns and the active involvement of medical professionals in awareness campaigns can be effective. Moreover, increasing medical students’ knowledge of HPV infections and vaccines can enhance the general public’s awareness of HPV screening and prevention techniques. It is recommended to replicate the current study in a different setting with a larger population and assess how educational interventions impact women’s awareness and attitudes toward cervical cancer. To increase HPV vaccination rates and decrease the incidence of cervical cancer, community advocacy initiatives and strategic planning must be developed and implemented in Saudi Arabia.

## Figures and Tables

**Figure 1 vaccines-11-01188-f001:**
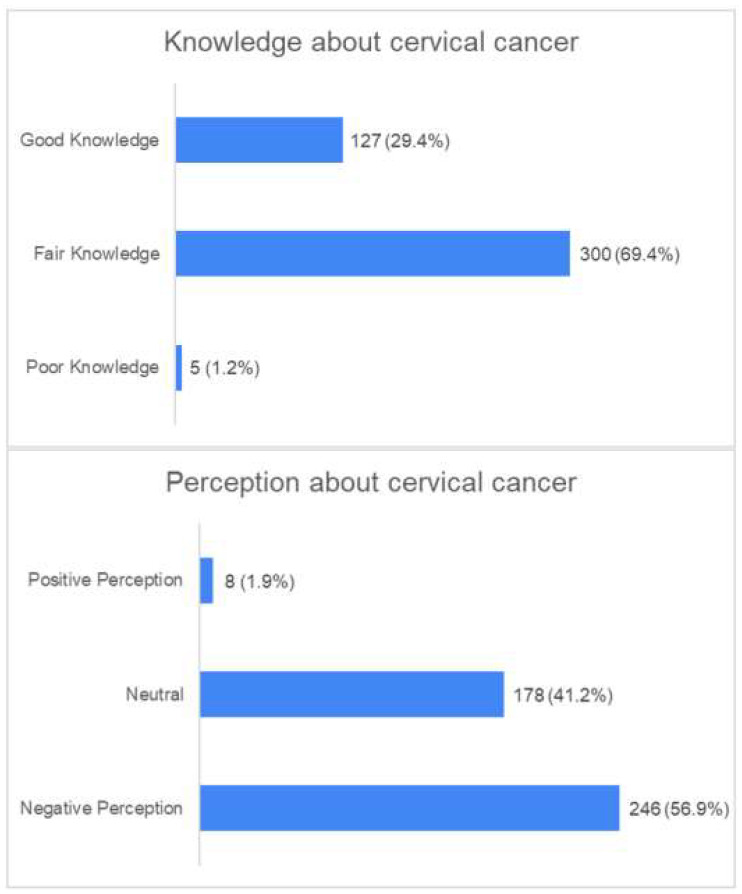
The percentages of knowledge and perception levels among the participants.

**Figure 2 vaccines-11-01188-f002:**
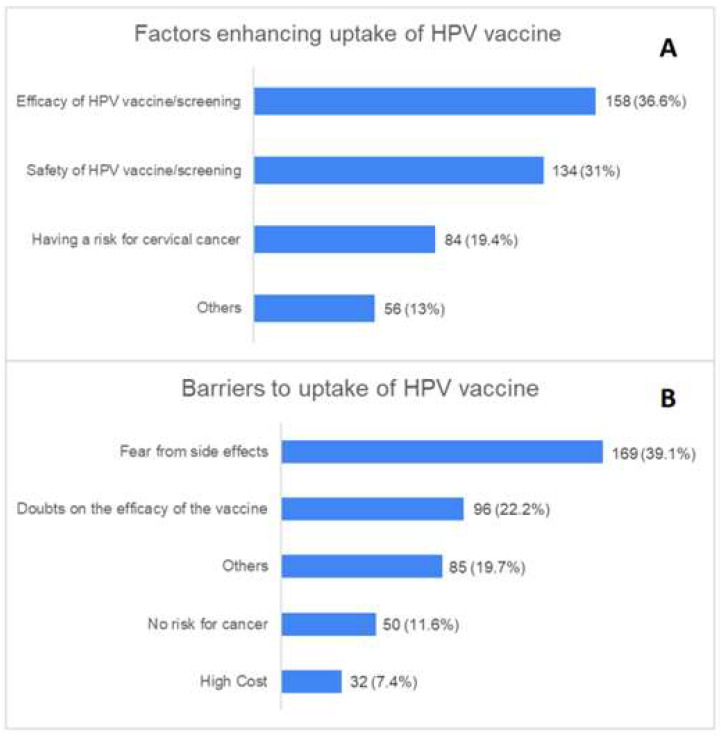
The percentages of participants’ responses regarding the facilitators (**A**) and barriers (**B**) toward the vaccine.

**Figure 3 vaccines-11-01188-f003:**
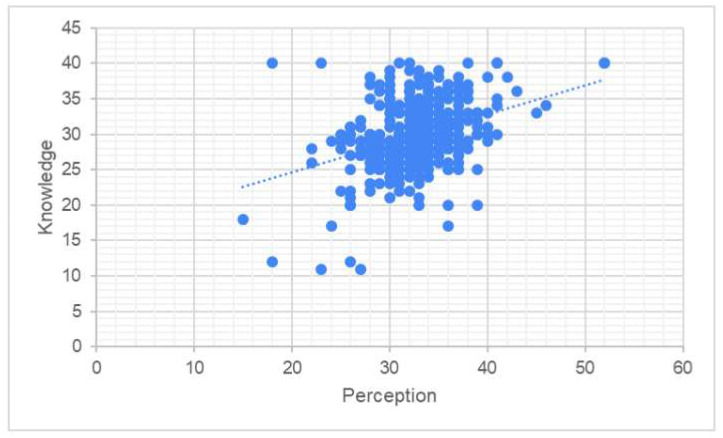
Correlation between knowledge and perception.

**Figure 4 vaccines-11-01188-f004:**
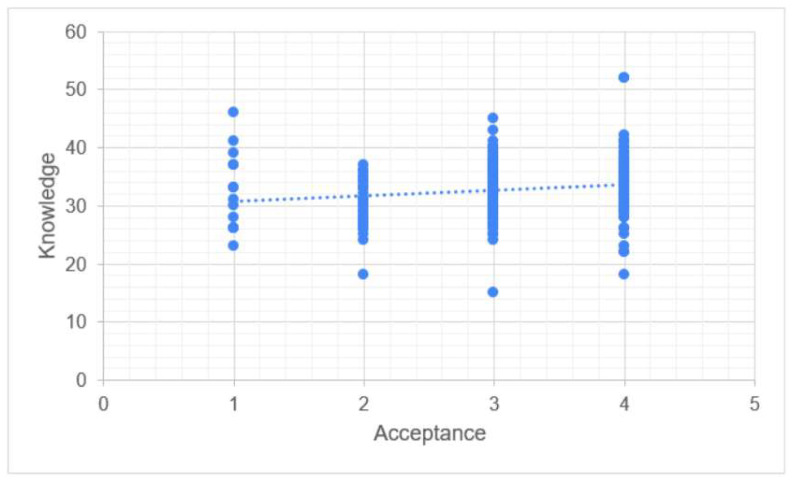
Correlation between knowledge and acceptance.

**Figure 5 vaccines-11-01188-f005:**
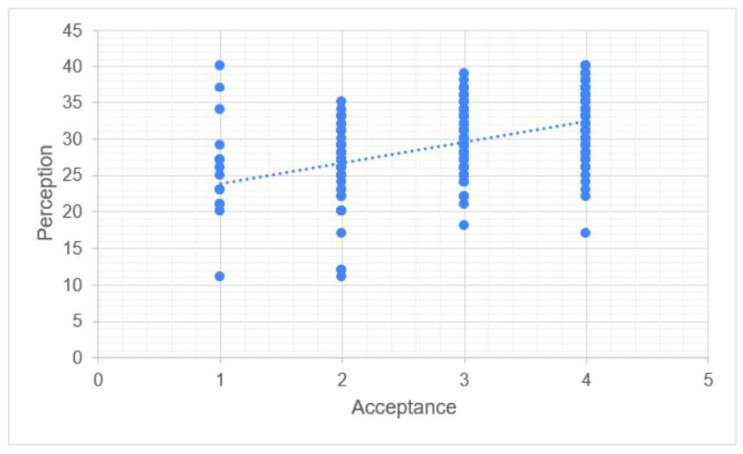
Correlation between perception and acceptance.

**Table 1 vaccines-11-01188-t001:** Demographic and cancer-related characteristics (*n* = 421).

Parameter	Category	*n* (%)
Age	18 to <30 y	218 (51.8%)
	30 to <40 y	105 (24.9%)
	40 to <50 y	98 (23.3%)
Marital status	Single	169 (40.1%)
	Married	231 (54.9%)
	Widow	4 (1.0%)
	Divorced	17 (4.0%)
Education	School education	101 (24%)
	Bachelor degree	273 (64.8%)
	Diploma degree	39 (9.3%)
	Postgraduate degree	8 (1.9%)
Employment status	Student	158 (37.5%)
	Employed	86 (20.4%)
	Unemployed	177 (42.1%)
Region *	Northern Region	163 (38.7%)
	Southern Region	135 (32.1%)
	Eastern Region	19 (4.5%)
	Western Region	50 (11.9%)
	Central Region	54 (12.8%)
	Have a family history of cancer	114 (27.2%)
Cancer-related characteristics	Ever heard about cervical cancer	332 (78.9%)
	Ever heard about HPV vaccination	131 (31.1%)
	Ever heard about screening	276 (65.6%)
	Ever screened against CC	35 (8.3%)
	Ever vaccinated against CC	46 (10.9%)
	Having family who have been screened	85 (20.2%)
	Having family who have been vaccinated	53 (12.6%)

* The record had 2 missing values.

**Table 2 vaccines-11-01188-t002:** Description of the scores of participants’ knowledge and perceptions (*n* = 421).

Parameter	Knowledge	Perception
Minimum, Maximum	20.0, 46.0	12.0, 44.0
Mean ± SD	34.7 ± 3.7	33.1 ± 5.0
Median (IQR)	35.0 (33.0, 37.0)	33.0 (30.0, 36.0)

**Table 3 vaccines-11-01188-t003:** Results of the regression analysis of participants’ knowledge (*n* = 421). * = statistically significant.

Parameter	Category	Univariate		
		Slope	Intercept	*p*
Age	18 to <30 y	-	-	
	30 to <40 y	0	33	0.302
	40 to <50 y	0	33	0.747
Marital status	Single	-	-	
	Married	0	33	0.48
	Widow *	1	33	<0.001
	Divorced *	1	33	<0.001
Education	School education	-	-	
	Bachelor degree	0	33	0.524
	Diploma degree	0	33	0.776
	Postgraduate degree *	−3	33	<0.001
Employment status	Student	-	-	
	Unemployed	0	33	0.643
	Employed	0	33	0.771
Region	Northern Region	-	-	
	Southern Region	0	33	0.535
	Eastern Region *	−1	33	<0.001
	Western Region *	1	33	<0.001
	Central Region *	−1	33	<0.001
Have a family history of cancer *	No	-	-	
	Yes	−1	33	<0.001
Ever heard about cervical cancer *	No	-	-	
	Yes	1	33	<0.001
Ever heard about HPV vaccination	No	-	-	
	Yes	0	33	0.169
Ever heard about screening *	No	-	-	
	Yes	1	32	<0.001
Ever screened against CC	No	-	-	
	Yes	0	33	0.451
Ever vaccinated against CC *	No	-	-	
	Yes	1	33	<0.001
Having family who have been screened *	No	-	-	
	Yes	1	33	<0.001
Having family who have been vaccinated	No	-	-	
	Yes	0	33	0.238

**Table 4 vaccines-11-01188-t004:** Results of the regression analysis of participants’ perceptions (*n* = 421).

Parameter	Category	Univariate		
		Slope	Intercept	*p*
Age	18 to <30 y	-	-	
	30 to <40 y	−1	30	<0.001
	40 to <50 y	1	29	0.006
Marital status	Single	-	-	
	Married	1	29	0.002
	Widow	−2	29	<0.001
	Divorced	−1	29	<0.001
Education	School education	-	-	
	Bachelor degree	−1	30	0.007
	Diploma degree	1	29	0.705
	Postgraduate degree	−1	29	<0.001
Employment status	Student	-	-	
	Unemployed	−1	30	<0.001
	Employed	0	29	0.009
Region	Northern Region	-	-	
	Southern Region	1	29	0.012
	Eastern Region	0	29	<0.001
	Western Region	1	29	0.244
	Central Region	−1	30	<0.001
Have a family history of cancer	No	-	-	
	Yes	−0.5	29.5	0.003
Ever heard about cervical cancer	No	-	-	
	Yes	1	29	<0.001
Ever heard about HPV vaccination	No	-	-	
	Yes	1	29	<0.001
Ever heard about screening	No	-	-	
	Yes	0.5	29	<0.001
Ever screened against CC	No	-	-	
	Yes	1	29	0.604
Ever vaccinated against CC	No	-	-	
	Yes	1	29	0.121
Having family who have been screened	No	-	-	
	Yes	1	29	0.028
Having family who have been vaccinated	No	-	-	
	Yes	1	29	0.281

**Table 5 vaccines-11-01188-t005:** Correlation between knowledge, perception, and acceptance (*n* = 421).

		Knowledge	Perception	Accepts to Take HPV Vaccine/Screening
Knowledge	Spearman Correlation	1	0.318	0.177
	*p*-value		0.000	0.000
Perception	Spearman Correlation	0.318	1	0.476
	*p*-value	0.000		0.000
Accepts to take HPV vaccine/screening	Spearman Correlation	0.177	0.476	1
	*p*-value	0.000	0.000	

## Data Availability

All data are available upon contact with the corresponding author.

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
