# Peer review of "Knowledge, Perception, and Acceptance of HPV Vaccination and Screening for Cervical Cancer among Saudi Females: A Cross-Sectional Study"

_vaccines, 2023, doi:10.3390/vaccines11071188_

Round 1

Reviewer 1 Report

I was invited to revise the paper entitled "Knowledge, Perception, and Acceptance of HPV Vaccination and Screening for Cervical Cancer among Saudi Females: a cross-sectional Study". It was a cross-sectional study aimed to evaluate attitudes  and associated factors towards HPV vaccination and cervical screening among  women from Saudi Arabia.

The topic is interesting and poor studies were performed in SA on this topic.

The paper needs several improvements:

- Statistical analysis section was poor and methods are not appropriare. Score variables are discrete variables and should be presented as median and IQR. In addition, they were not normally distributed so linear regression analysis was not appropriate;

- Pearson correlation was not appropriate for this nature of data;

- Education level and employment variables are too fragmentated. Due to the small sample size, I suggest to decrease the number of categories for these variables;

- In introduction section, Authors should better describe the vaccination schedule of HPV in SA and cervical cancer screening proposed;

- In Table 1 Authors should also report data divided by women with poor knowledge and women with high knowledge;

- Among discussions, Authors should compare their results with similar studies from other countries;

- In addition, it is important to highlight how HPV was related to other kind of cancer than cervical cancer.

Author Response

Regards. 

Reviewer 2 Report

The manuscript (ID: vaccines-2439266) aimed to assess women's knowledge, perceptions, and acceptance concerning HPV vaccination and cervical cancer screening in Saudi Arabia, as well as the contributing variables to women's screening and vaccine acceptability.

But, some issues in this manuscript require major revision (the Introduction and Methods sections):

  • Lines 12-13: Specify the year.
  • Line 26: Specify whether it is incidence or mortality. It says `global level`? Cite the appropriate reference for the statement in this sentence.
  • Lines 29-32: Is reference No. 4 cited for the data presented in these 2 sentences? Check and correct this. Reference No. 4 in the list of References refers to China, but does not mention Saudi Arabia at all. Explain.
  • Line 36: `pep test`? Correct this.
  • Line 43: Insert a new paragraph in which the following should be described in detail:
    • Whether and when an organized / population / national screening for cervical cancer was implemented in Saudi Arabia, what age of women is planned for cervical screening, what screening tests are planned for cervical screening in Saudi Arabia?
    • Is the HPV vaccine available in Saudi Arabia and since when? Is the HPV vaccination implemented in the schedule of mandatory vaccinations, is the HPV vaccine provided for both women and men in Saudi Arabia, for what age is the HPV vaccination provided? Are there any other indications for HPV vaccination in Saudi Arabia?
  • Lines 51-56: Repetition of almost the same sentences. Correct this.
  • Lines 57-117: The paper uses the appropriate methodology, which is described in detail. But the following corrections need to be made:
    • Describe the Study population in more detail.
    • State the criteria for inclusion in this study.
    • Specify the exclusion criteria in this study.
    • Does `region` mean the region in Saudi Arabia? Explain this.
  • Lines 183-274: In the Discussion section, a comparison of the results of this study with the results of similar studies in other populations was carried out in a correct manner. In addition, satisfactory explanations for differences in results between studies were provided. The discussion is supported by citing appropriate references.
  • Lines 282-285: In Table 1 it is presented that the respondents were from several regions in Saudi Arabia? Explain this text in the subsection about limitations: was there one city or were women from the entire country and all regions included in this study? Notes:
    • It is necessary to improve the discussion of limitations, of which there are not a few in this work.
  • Lines 297-311: The paragraph titled `Recommendation and implications` is very useful.   

The quality of English language is appropriate. 

Author Response

Please the attachment.

Regards. 

Round 2

Reviewer 1 Report

I was invited to review the revised version of the paperentitled "Knowledge, Perception, and Acceptance of HPV Vaccination and Screening for Cervical Cancer among Saudi Females: a cross-sectional Study". Authors addressed the great part of previous cooments.

observations:

- Lines 35-36 require referencies;

- Table 3 requires multiple comparisons correction (bonferroni? false discovery rate?);

- regression coefficent should be reported with relative 95% confidence interval;

- Authors reported to perform Spearman correlation so they should remove R coefficients and replace it with Rho troghout the paper;

Author Response

Regards.

Reviewer 2 Report

The authors adequately addressed all my comments from the previous round of peer review.

The corrections made in the revised version of the paper significantly improved the informativeness and readability of the paper.

Thanks to the authors.  

The quality of English language is appropriate. 

Author Response

Thank you.

Round 3

Reviewer 1 Report

Minor revisions:

- Reference 5 is too old. Replace it with more recent literature. Several studies were published recently on this topic;

- CI of regression coefficient can be calculated.

Author Response

Reviewer comment: Reference 5 is too old. Replace it with more recent literature. Several studies were published recently on this topic.

Author response: new reference added (2021)

Reviewer comment: - CI of regression coefficient can be calculated.

Author response:  Thanks for the comment. We performed this analysis using RStudio Software, we searched the literature for available R codes that can estimate the 95% CI and were not successful. Kindly inform us how to calculate the CI and we will pleasantly do it.

Regards.